# Innate Immune Recognition, Integrated Stress Response, Infection, and Tumorigenesis

**DOI:** 10.3390/biology12040499

**Published:** 2023-03-25

**Authors:** Klara Kubelkova, Vanda Bostik, Lokesh Joshi, Ales Macela

**Affiliations:** 1Department of Molecular Pathology and Biology, Faculty of Military Health Sciences, University of Defence, 500 01 Hradec Kralove, Czech Republic; 2Department of Epidemiology, Faculty of Military Health Sciences, University of Defence, 500 01 Hradec Kralove, Czech Republic; 3Glycoscience Group, National Centre for Biomedical Engineering Science, University of Galway, H91 W2TY Galway, Ireland

**Keywords:** innate immunity, infection, tumorigenesis

## Abstract

**Simple Summary:**

The cellular and humoral mechanisms of natural immunity are at the beginning of all the defence processes of an organism. Their defence function is based on the ability to recognize pathogen-associated molecular patterns and damage (danger)-associated molecular patterns using pattern recognition receptors. Recognition of pathogen-associated molecular patterns and/or damage (danger)-associated molecular patterns has a crucial role in the activation of the innate immune defence mechanisms. An inflammatory response, as a basic event in early defence occurs, which is the complex, developmentally acquired ability of living organisms to react to various damages. If it is not strictly regulated, it turns into damaging inflammation. This is common both in some infectious diseases, such as prolonged COVID-19, and in precancerous tissue preceding the development of tumors. This short review presents a vision of an integration of innate immune recognition, cell-autonomous stress response, infection, and tumorigenesis.

**Abstract:**

Engagement of PRRs in recognition of PAMPs or DAMPs is one of the processes that initiates cellular stress. These sensors are involved in signaling pathways leading to induction of innate immune processes. Signaling initiated by PRRs is associated with the activation of MyD88-dependent signaling pathways and myddosome formation. MyD88 downstream signaling depends upon the context of signaling initiation, the cell (sub)type and the microenvironment of signal initiation. Recognition of PAMPs or DAMPs through PRRs activates the cellular autonomous defence mechanism, which orchestrates the cell responses to resolve specific insults at the single cell level. In general, stressed endoplasmic reticulum is directly linked with the induction of autophagy and initiation of mitochondrial stress. These processes are regulated by the release of Ca^2+^ from ER stores accepted by mitochondria, which respond through membrane depolarization and the production of reactive oxygen species generating signals leading to inflammasome activation. In parallel, signaling from PRRs initiates the accumulation of misfolded or inappropriately post-translationally modified proteins in the ER and triggers a group of conserved emergency rescue pathways known as unfolded protein response. The cell-autonomous effector mechanisms have evolutionarily ancient roots and were gradually specialized for the defence of specific cell (sub)types. All of these processes are common to the innate immune recognition of microbial pathogens and tumorigenesis as well. PRRs are active in both cases. Downstream are activated signaling pathways initiated by myddosomes, translated by the cellular autonomous defence mechanism, and finalized by inflammasomes.

## 1. Introduction

The fundamental properties of living matter include continuously creating diversity and occupying any space that is available, including such space that is already occupied by other living matter. This property is in fact the fundamental condition leading to the creation of the eukaryotic cell, which has a symbiotic origin. In order to survive, biological systems must also have other properties in addition to the aforementioned properties. They must be able to discriminate between self and non-self and protect their identity in order to develop higher functions. Thus, through gradual evolution, a number of genes encoding receptor structures have become fixed even though evolution is a dynamic process. They are able to recognize both the evolutionarily conserved structural patterns characterizing the non-self and the patterns that are indeed self but that do not meet the criteria of functionality [1]. 

## 2. The Interplay of TLRs and Complement Receptors Reinforces Innate Immunity

Multicellular biological species, including humans, have developed defensive systems based on either innate or adaptive immunity. The innate immune system is such a basic defence system meeting the requirements of both discriminating self from non-self and the ability to identify sequentially, spatially, or functionally pathological molecular structures. The innate immune system recognizes pathogen-associated molecular patterns (PAMPs) as well as damage-associated molecular patterns (DAMPs). These patterns are recognized by pattern recognition receptors (PRRs) distributed on cell surfaces and also by receptors located in the cytosol and cellular organelles [2]. Most PAMPs and DAMPs serve as so-called ‘Signal 0s’ that bind appropriate PRRs [3]. PRRs recognize their ligands as monomeric structures or form homo- or heterodimers with receptors of the same group (Table 1). During the initiation of the innate immune response and during the regulation of the adaptive immune response, the PRRs can collaborate with receptors of an unrelated group. The receptors from the RIG-I-like receptors group can create cross-talk with Toll-like receptors (TLRs) to initiate type 1 interferon and pro-inflammatory cytokines production. Not all receptors, however, are constitutively expressed on all cell types of an organism.

The interplay of TLRs and complement receptors (CRs) reinforces innate immunity, thus regulating the inflammatory reaction in a positive or negative sense, which can lead either to a weakening of the natural defence reaction or to a cycling of signaling that leads to damaging inflammation [4,5,6]. Interrupting the looped chain of signals leading from collaborating TLRs and including CRs can be used in clinical practice. Inhibition of the central complement molecules C3 and C5 and the CD14 molecule, which is a co-receptor for several TLRs, has been suggested as a “dual blockade” approach to regulate improper or uncontrolled innate immune activation threatening the host [7]. Such exaggerated responses of innate immune systems can occur during innate immune recognition of PAMPs (or DAMPs), and it can introduce disharmony of intracellular signaling, affect stress reactions of cell organelles (endoplasmic reticulum and mitochondria), and affect cell-autonomous host defence. 

## 3. Cell-Autonomous Host Defense, Inflammation, and Tumorigenesis

Cell-surface recognition of PAMPs by PRRs initiates cellular stresses, which frequently are accompanied by accumulation of misfolded or inappropriately post-translationally modified proteins in the endoplasmic reticulum (ER). Such unwanted accumulation of misfolded proteins triggers a conserved emergency rescue pathway known as unfolded protein response (UPR), which is another stressor for cell organelles [8,9,10]. Stressed ER induces release of Ca^2+^ from ER stores, which is accepted by mitochondria and signals to mitochondria to produce reactive oxygen species (mtROS) leading to inflammasome activation [2]. Production of mtROS as a consequence of PAMPs recognition by PRRs and downstream signaling cascades creates conditions for genetic information damage and for microenvironment suiting uncontrolled proliferation of tumor-transformed cells. Interconnection among the innate immune recognition on the cell membrane by PRRs and cell-autonomous defence is a complex process leading to stress response of cellular organelles, both endoplasmatic reticulum and mitochondria. Whole process generates the secondary signals in the form mitochondrial DAMPs [11], which, finally, generate functional profile of internal cell-autonomous defence (Figure 1).

Thus, pathogen recognition by PRRs is the key to enabling the host cell to detect the presence and precise location of a pathogen, as well as the primary signal for initiating cell-autonomous host defence, inducing innate immune responses or, as an unwanted result, initiating a tumorigenesis process. These first events are not exclusively a matter of TLRs. Opsonization of pathogens with fresh serum or purified antibodies redirects the interaction of bacteria with host cells from TLRs or mannose receptors to CRs and Fc-gamma receptors (FcγRs). Receptors such as CRs or FcγR also engage in modulating intracellular signaling pathways and activating cell-autonomous defence. The signaling by the C5a receptor upon activation generates pro-survival and anti-apoptotic responses. The C5a binding to C5aR decreases apoptosis in several cell types, including colon cancer cell lines. Activation of C5aR appears to be a double-edged sword. On the one hand, C5aR activation generates pro-survival signals by decreasing apoptosis in neutrophils and T cells, which may benefit intracellular pathogen proliferation. On the other hand, it increases the proliferation of endothelial cells and colon cancer cell lines, an effect that has a relevant role in carcinogenesis. The pro-cancerous role of C5a binding to C5aR has been demonstrated in murine models of cervical and ovarian cancers [12]. The consequence of C5aR cooperation with TLRs activated by pathogens is the production of pro-inflammatory cytokines IL-6 and IL-1β from mononuclear cells which participate in the accumulation and activation of myeloid-derived suppressor cells (MDSCs) within tumors [13]. Moreover, activation of the C5a receptors results in increased production of reactive oxygen and nitrogen species, which are known mediators that suppress CD8+ T cell functions. Thus, the uncontrolled inflammation associated with PAMPs recognition forced by PRR cooperation seems to be the dominant factor enabling the start of the tumorigenesis process and cancer progression [14,15].

## 4. Innate Immune Defence and Tumorigenesis

DAMPs are molecules released after an attack of mental or physical stress, tissue injury, and/or cellular stress from damaged or dying cells. DAMPs are also released by a wide range of tumors [16]. The S100s proteins, histones, amphoterin (HMGB1), or HSPs secreted by tumor cells are recognized by PRRs as DAMPs [17]. Generally, the S100s proteins are recognized by TLRs or RAGE, the receptor for advanced glycation end-products [18]. HMGB1 interacts with RAGE and several TLRs (TLR2, TLR4, and TLR9) depending on the cell types [19], the histones react with TLR2, TLR4, or TLR9, histone H4 is recognized by the TLR4/MD2 heterodimer, and the nucleosome, which is the histone-DNA complex, is the TLR9-specific ligand [20]. The recognition of tumor-derived DAMPs therefore usually initiates the activation of signaling cascades, thus leading to sterile inflammation, which forms a reinforcing loop of tumorigenesis. Mitochondria also constitute a substantial source of intracellular DAMPs. They are represented by mitochondrial DNA, ATP, mitochondrial transcription factor A (TFAM), N-formyl peptides, succinate, and cardiolipin [11]. The initiation of intracellular signaling by the interaction of DAMPs with the respective receptors activates constitutive cell-autonomous immunity, representing the primordial innate defence and its functional expression [21]. Downstream of the PRRs ligated by DAMPs, there is activation of a shared set of signaling pathways represented by NF-κB, p38, and/or ERK. Their activation terminates in inflammasome assembly and the secretion of a range of cytokines represented by IL-1β, IL-18, IL-6, TNF, LT-β, IFNγ, and TGF-β, along with processes leading to inflammasome activation and control of autophagic processes. In parallel, deregulated mTOR signaling, for example by oxidative stress as a consequence of activated cell-autonomous immunity, is implicated in cell proliferation and is characteristic of various types of disorders, such as cancer, diabetes, and some neurodegenerative diseases [22]. Upstream signaling, regulating mTOR activity, corresponds to the PI3/AKT pathway, MAPK/ERK, JNK, or AMPK pathways, and the Wnt pathway. All these pathways respond to oxidative stress including from mtROS which has been shown to be an atypical activator of the AMPK signaling pathway. Downstream signaling occurs through a translation repressor protein 4E-BP1 and ribosomal S6 kinase S6K. The multiple pathways allow mTORC1 to inhibit autophagy while at the same time stimulating protein synthesis and cell growth; both can accumulate the damaged proteins and organelles and generate new DAMPs. The DAMPs, by promoting pathological chronic inflammatory reactions, constitute a complex risk factor for tumor progression. DAMPs can stimulate the process of tumorigenesis and promote tumor growth. Lymphokines IL-1, IL-6, and lymphotoxin have been identified as promoters of carcinogenesis. DAMPs, such as S100 proteins or heat shock proteins, activate inflammatory pathways and release IL-1, IL-6, LT-β, IFN-γ, TNF, and TGF-β. Similarly, ATP as a DAMP, or adenosine itself, or IL-1α induce signals promoting carcinogenesis by intensifying processes leading to inflammation, immunosuppression, angiogenesis, and tumor cell proliferation. In this context, DAMPs appear to enhance tumor development in the early stages of tumorigenesis [23].

Interaction of general stressors in the form(s) of PAMPs and/or DAMPs with PRRs, initiates a process of innate immune recognition and ensuing activation of a signaling cascade that subsequently leads to cellular stress. Stress is a key event in activating cell-autonomous defence terminating in the integrated stress response (ISR), an evolutionarily conserved intracellular signaling network. The ISR terminates in either cellular homeostasis or cell death. The stresses, regardless of whether they are of extracellular or intracellular origin, are sensed by four specialized kinases (PERK, GCN2, PKR, and HRI) that activate the eukaryotic translation initiation factor eIF2. That factor, among other specific mRNAs, triggers the translation of the key activating transcription factor ATF4 which is a multifunctional transcription regulatory protein that participates in a variety of cellular responses to different stresses or intercellular signaling molecules such as, for example, growth factors [24]. During their interaction with host cells, both bacterial and viral pathogens modulate the ISR to ensure replication of self [25,26]. Even subtle modification of these processes by microorganisms can lead to translation reprogramming and the initiation of tumorigenesis. Moreover, the tumor cell, if already present, is capable of manipulating the ISR, and the ISR kinases (PKR, PERK, and GCN2) have been shown to be implicated in cancer cell proliferation [27,28].

## 5. Intracellular Signaling, Infection, and Tumorigenesis 

From the viewpoint of intracellular signaling, DAMPs and PAMPs both activate a similar spectrum of signaling pathways. Inflammasomes activation can be the common feature of both the induction of innate immune response and tumorigenesis. Assembly and activation of the NLRP3 inflammasome are possible by a variety of microbial PAMPs, for example by pneumolysin (*Streptococcus pneumoniae*), hemolysis (*Staphylococcus aureus*), flagellin (*Salmonella typhimurium*), or RNA (*Escherichia coli*), but also in the microenvironment of breast cancer, lung cancer, or melanoma. Moreover, NLRC4 is activated by *Mycobacterium tuberculosis* and *S. typhimurium*, as well as in gastric cancer. Activation of the AIM2 inflammasome has been proven during *Francisella tularensis* infection; in other infections involving intracellular bacteria such as *Listeria*, *Brucella*, or *Legionella*; and similarly in breast cancer and lung cancer summarized in [29,30,31]. 

It is widely accepted that certain bacterial or viral infections are associated with the induction of processes leading to tumor formation. Among bacteria, significant examples include the association of *Helicobacter pylori* and gastric adenocarcinoma or mucosa-associated lymphoid tissue lymphoma [32], *Streptococcus bovis* and colon cancer [33], *Chlamydia pneumoniae* and lung cancer [34], *Fusobacterium nucleatum* and colorectal cancer [35], or *Prevotella melaninogenica* and oral cancer [36]. Viruses with oncogenic potential include, for example, the HTLV-1 virus from the *Retroviridae* family or the hepatitis B and C viruses, which are associated with the possible development of hepatocellular carcinoma, as well as the human papillomavirus which is associated with the development of cervical, vaginal, penile, or anal cancers [37]. Data from the 1990s indicate that viruses are etiologically linked to about 20% of human malignancies, and among those, approximately 15% of worldwide cancer incidence is associated with the papillomavirus, hepatitis B virus, Epstein–Barr virus, and human T cell leukemia–lymphoma. Viruses from the *Retroviridae* family are responsible for about 8% to 10% of virus-induced malignancies [38,39]. 

There currently is open discussion as to whether the SARS-CoV-2 virus can initiate processes leading to the formation of lung tumors. The inflammatory response to SARS-CoV-2 in lung tissues prompted by the production of pro-inflammatory cytokines such as TNF-α, IL-1β, IL-8, and IL-6 influences the lung microenvironment and modulates the coordination of innate immune reactions [40]. Such conditions are very similar to pre-cancerous conditions of lung tissue [41]. For this reason, therefore, it is possible to consider SARS-CoV-2 infection as a risk factor for the initiation of lung tissue tumorigenesis and, accordingly, it has been recommended to monitor patients who have recovered from COVID-19 for the possible development of lung tumors [42,43].

In general, there are essentially three ways by which an infectious process could turn into carcinogenesis. In the first case, some viruses directly insert their genes into the cell genome, thereby causing the cell to go out of control (e.g., human T-lymphotropic virus-1). Second, long-term inflammation induced by infection can lead to changes in affected and surrounding cells and these changes can induce tumor transformation (e.g., human papillomaviruses, the Epstein–Barr virus, the hepatitis B virus, and the hepatitis C virus; bacteria such as *Helicobacter pylori* and *Chlamydia trachomatis*, and also parasites such as *Opisthorchis viverrini*, *Clonorchis sinensis*, or *Schistosoma haematobium*). Third, carcinogenesis can be due directly to the weakening of the infected organism’s immune system (e.g., human immunodeficiency virus). 

There also are microorganisms, however, that downregulate the process of tumorigenesis even if they do activate some type of inflammasome. An example of a bacterium appearing to possess anti-tumor properties is *Listeria monocytogenes*, which activates several inflammasomes (AIM2, NLRP1B, NLRP3, NLRP6, and NVRC4) by its PAMPs, flagellin, lipoteichoic acid, listeriolysin-O, or dsDNA and triggers effective innate immune response and apoptotic death of cancer cells [44,45,46]. Along with the pleiotropic effects of Listeria in anti-tumor immunity, *L. monocytogenes* has also been tested as a vector of target antigens for tumor immunotherapy [47]. The application of mycobacteria, such as to produce the Bacillus Calmette–Guerin (BCG) vaccine, in the treatment of various malignancies, including intralesional BCG immunotherapy, has been demonstrated to offer a highly efficient and cost-effective remedy for cancer treatment [48,49,50]. It should be noted, however, that the first attempts to treat tumors using bacteria or their products had already been made at the end of the 19th century. In the 1880s, William Coley, a New York City surgeon, observed improvement in tumor patients after acute bacterial infection [51,52]. Subsequently, on the basis of the data he found in the older literature, Coley used bacteria or their toxins for the treatment of tumors in advanced stages in order to induce a violent thermal response in patients [47]. Finally, in collaboration with Robert Koch, a mix of bacterial toxins was developed for use in the treatment of cancer, and these are now referred to as ‘‘Coley’s toxins’’ [52,53].

## 6. Conclusions

To summarize this mini-review, we would like to draw attention to and stimulate ensuing discussions regarding the early stages of infection processes, which lead to modulation of signaling pathways providing, on the one hand, a basis for inducing powerful adaptive specific immunity against infection while, on the other hand, preventing disharmony in the fine regulation of innate immune responses after innate immune recognition of PAMPs or DAMPs. Disharmony in intracellular and intercellular signaling can lead to a disproportionate inflammatory response, thereby creating a microenvironment for the formation of pre-cancerous conditions in infected and inflamed tissues. We believe that such a discussion could be very beneficial, given that protective immunity and damaging inflammation leading to pre-cancerous conditions share important signaling and executive elements such as innate immune recognition of PAMPs or DAMPs; an integrated stress response; inflammasome assembly and activation; as well as mitophagy, autophagy, or processes of apoptosis. We nevertheless lack sufficient basic knowledge essential to fully disclose and identify early processes ongoing during the onset of disease, including their modulation by microbial agents or already existing tumor cells.

## Figures and Tables

**Figure 1 biology-12-00499-f001:**
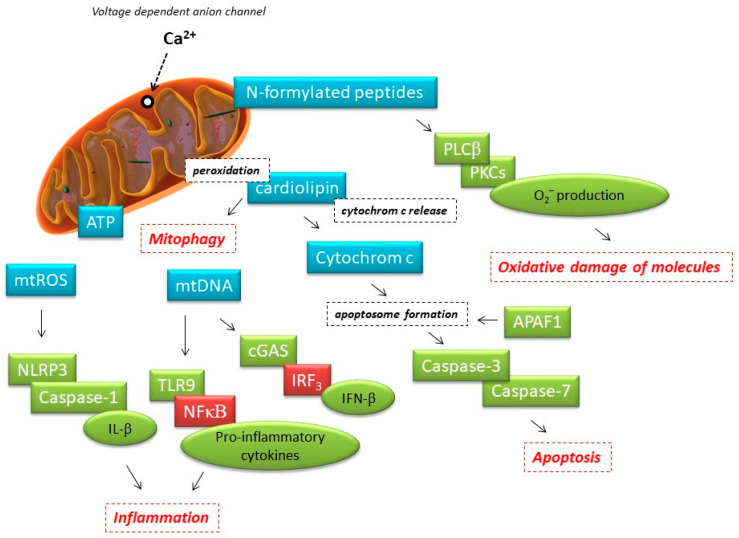
The initiation of intracellular signaling by the interaction of DAMPs or PAMPs with the respective receptors activates constitutive cell-autonomous immunity, representing the primordial innate defence and its functional expression. The internal cell response is initiated by the release of Ca^2+^ from the stressed ER that is accepted by mitochondria and initiates the generation of secondary stress signals. They are represented by mtROS, N-formyl peptides, and cardiolipin, as well as cytochrome c released from mitochondria, which finalized the effector mechanisms of the autonomous stress response. Another mitochondrial-damage-associated molecular pattern, ATP, initiates the assembly of the inflammasome(s) and initiates inflammation or autophagy, with dependency on the stressor(s).

**Table 1 biology-12-00499-t001:** Pattern recognition receptors and their ligands according to their relationships to subcellular structures.

Subcellular Structures	PRRs	Ligand(s)
** *Plasma membrane* **	TLR1/TLR2	Bacterial lipoproteins and unconventional lipopolysaccharides (LPS)
	TLR2/TLR2	Peptidoglycan and zymosan
	TLR2/TLR6	Mycobacterial lipoproteins
	TLR2/TLR10 *	Peptidoglycan and (triacyl) lipopeptides
	TLR4	Conventional (enterobacterial) LPS
	TLR5	Flagellin
	TLR10/TLR10 *	HIV-gp41 and diacylated lipopeptides
	TLR11 ** (mouse, rat)	Profilin
	Dectin 1	β glucan
	Dectin 2	α mannan
	Mannose receptor	n-linked mannan
	DC-SIGN	Mannose-containing glycoproteins, ICAM2, and ICAM3
	Mincle	Glycolipids, trehalose-6,6’-dimycolate, and cord factor
	Mannan-binding lectin	Carbohydrates and senescent and apoptotic cells
	Gelectin 3	Beta-galactosidase
** *Endosomes* **	TLR3	Double-stranded RNA
	TLR7	Single-stranded RNA
	TLR8	Single-stranded RNA
	TLR9	CpG-DNA
** *Cytosol* **	NOD1	D-gamma-Glu-mDAP
	NOD2	Muramyl dipeptide
	NLRP3	Necrotic cells, uric acid, ATP, biglycan, and hyaluronan
	NLRP4	Flagellin
	AIM2	Double-stranded DNA
	RIG1	ssRNA and short blunt dsRNA
	MDA5	Long dsRNA
	LGP2	Double-stranded RNA
** *Endoplasmic reticulum* **	STING	Cytosolic DNA

Notes: This table was compiled from data presented in generally accessible original publications relating to individual receptors. Due to the general nature of the information presented in the table, these publications are not listed in the References section. * TLR10 is a pseudogene in mice. All other mammalian species contain an intact copy of the TLR10 gene. ** TLR11 in mice and rats is encoded by the intact gene. In humans, it is represented only by a pseudogene.

## Data Availability

Not applicable.

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
