# Peer review of "Innate Immune Recognition, Integrated Stress Response, Infection, and Tumorigenesis"

_biology, 2023, doi:10.3390/biology12040499_

Round 1

Reviewer 1 Report

In this interesting opinion article, Kubelkova et al. deal with the function of PAMPs and DAMPs in cancer progression, summarizing findings from the more recent reports that describe an important role for cellular intrinsic mechanisms, infections and organized innate immune responses in tumorigenesis.

In general, the manuscript is well written, but more details of the role played by PAMPs and DAMPs in tumorigenesis should be incorporated. Moreover, important references are missing.

Specific points:

1.- The reference by Tang et al.,PAMPs and DAMPs: signal 0s that spur autophagy and immunity” (Immunol Rev. 2012. 249:158-75. doi: 10.1111/j.1600-065X.2012.01146.x) should be quoted next the reference no. 2 at the manuscript’s Introduction section.

2.- One of the weaknesses of the manuscript is that the importance of DAMPs from tumour microenvironment and tumour-derived proteins in cancer progression is not sufficiently treated. The authors should include a brief discussion of this topic and include also the reference by Jang et al. “Interactions between tumour-derived proteins and Toll-like receptors” (Exp. Mol. Med. 2020. 52:1926–1935. doi.org/10.1038/s12276-020-00540-4).

3.- Table 1: Besides ICAM-3, another important ligand for DC-SIGN is ICAM-2 (Geijtenbeek, et al. “DC-SIGN–ICAM-2 interaction mediates dendritic cell trafficking” Nat. Immunol., 2000. 1: 353-357. doi: 10.1038/79815).

4.- Something seems wrong with the reference to Figure 1 on page 4, line 97. This figure does not show the fostered entry of pathogens into the host. The figure would be more complete and accurate if a detailed representation of the upstream PAMPs, DAMPs and PRRs inducing Ca2+ release from the ER were depicted. Moreover, do the authors consider that cytochrome C, cardiolipin and N-formylated peptides released from mitochondria are DAMPs?

5.- Page 5, lines 133-134: May the authors specify which DAMPs are sourced from mitochondria?  

Reviewer 2 Report

This work is devoted to the relationship between innate immune recognition, integrated stress response, infection and tumorigenesis. The relevance of this article is not in doubt, since the response to infection and damaging inflammation leading to precancerous conditions are mediated through common signaling elements. The authors described in detail the interaction between TLRs and complement receptors leading to enhanced immune response. They illustrated the article with examples of microbial PAPMs that activate inflammasomes, which may be a common feature of both induction of the innate immune response and tumorigenesis. Therefore, this work opens new discussions for further research.

Small remarks and questions:

1. Lines 43-46: In the introduction, you write about recognizing patterns that characterize not-self and self that do not meet the criteria for functionality. At the same time, you did not indicate that the recognition of normally functioning self also occurs.

2. In the figure, «inflammation» is indicated twice. Also, for better perception, it is better to put two objects side by side and leave one «inflammation».

3. The C5a binding to C5aR decreases apoptosis in several cell types, including colon cancer cell lines (119-120 lines). And in what other types of cells their binding reduces apoptosis?

4. Lines 141-144: the interconnection between oxidative stress and deregulation of mTOR signaling is not described. The authors should expand this idea.

5. The chapter "Conclusions" is numbered, while other chapters were not numbered.

Round 2

Reviewer 1 Report

The authors have considerably improved the manuscript. The only issue pending is that the "Simple Summary" section sounds somewhat repetitive. The authors should improve it.

Author Response

Simple Summary has been modified.